# Development and Pilot Testing of Smartphone-Based Hearing Test Application

**DOI:** 10.3390/ijerph18115529

**Published:** 2021-05-21

**Authors:** Kashyap Patel, Linda Thibodeau, David McCullough, Emma Freeman, Issa Panahi

**Affiliations:** 1Department of Electrical and Computer Engineering, The University of Texas at Dallas, Richardson, TX 75080, USA; imp015000@utdallas.edu; 2School of Behavioral and Brain Sciences, The University of Texas at Dallas, Richardson, TX 75080, USA; thib@utdallas.edu (L.T.); david.mccullough2@utdallas.edu (D.M.); emma.freeman@utdallas.edu (E.F.)

**Keywords:** audiogram, hearing test, pure-tone audiometry, smartphone-based hearing test

## Abstract

Background: Identifying and treating hearing loss can help improve communication skills, which often leads to improved quality of life. Many people do not seek medical treatment and, therefore, go undiagnosed for an extended period before realizing they have hearing loss. This study presents a self-administered, low-cost, smartphone-based hearing test application (HearTest) to quantify the pure-tone hearing thresholds of a user. The HearTest application can be used with commercially available smartphone devices and an earphone with the mentioned specification. Methods: Air-conduction-based pure-tone audiometry for the smartphone application was designed and implemented to detect hearing thresholds using a traditional “10 dB down and 5 dB up” approach. Employed smartphone-earphone combination was calibrated with respect to a GSI-61 audiometer and insert earphone ER-3A to maintain clinical standards with the help of subjective testing on 20 normal-hearing (NH) participants. Results: Further subjective testing on 14 participants with NH and retesting on five participants showed that HearTest achieves high-accuracy audiogram within clinically acceptable limits (≤10 dB HL mean difference) when compared with the reference clinical audiometer. Hardware challenges and limitations in air-conduction-based hearing tests through smartphones and ways to improve their accuracy and reliability are discussed. Conclusion: The proposed smartphone application provides a simple, affordable, and reliable means for people to learn more about their hearing health without needing access to a formal clinical facility.

## 1. Introduction

Hearing loss affects 470 million people in the world and it is expected to double by the year 2050 [1]. Untreated hearing loss has a negative impact on cognitive abilities, social behavior, professional life, and memory. Additionally, it has been reported that only 16% of physicians’ screen for hearing loss in the USA. As hearing loss can be a gradual process, it is important to identify and provide options to enhance communication or identify if another issue is at hand, such as a pathology or neurological issue. People with hearing loss wait an average of seven years before seeking help [2]. In the case of age-related hearing loss, high-frequency sounds are typically lost first due to their location in the cochlea [3]. Friends, colleagues, or family members are often the first ones to notice a loved one’s loss in sound awareness. People are usually unaware of the fact that they have hearing loss and hesitate to visit a hearing-health professional, such as an audiologist, usually with an excuse of cost or time. To maintain health, regular hearing loss screening and treatment is necessary. If identified, hearing aids or cochlear implants can substantially help most people and early detection and treatment may facilitate adjustment to the new sound.

Many advanced adaptive signal processing algorithms have been developed that implement distinct parts of the hearing aid processing pipeline in modern smartphones in real time. The combination of such novel algorithms developed for monaural and binaural hearing applications and the smartphone provide an immensely powerful and cost-effective platform to assist people with hearing disorders [4,5,6,7,8,9,10]. A crucial part of the hearing aid processing pipeline is appropriate selection of fitting parameters (i.e., compression) based on the listener’s levels of hearing sensitivity. This is defined by the individual patient’s audiogram obtained in a clinical setting. As an important part of the smartphone-based platform, the hearing test algorithm proposed in this paper offers monaural and binaural hearing tests that would lead to appropriate automated selection of frequency dependent gain values and required parameters for the fitting algorithm.

Most people with moderate-to-profound hearing impairments live in low- and middle-income countries [11]. Conducting a professional hearing screening requires a clinic with audiometric services and an Audiologist. A formal clinical facility might not be physically available or affordable in underdeveloped or developing countries [12]. Having a widely acceptable and cost-effective solution to basic hearing tests is of the utmost importance. Even in well-developed countries, long test durations and commutes to the clinic may cause fatigue and can be problematic, especially for elderly people. There is a huge demand for mobile audiometry devices that can assist people to conveniently have their hearing health checked. Portable audiometric services do exist that can be used without reaching formal audiometric services within the community, but they require professionals to help interact with the portable audiometer [13]. An accurate, self-assisted, and portable solution to basic hearing tests is highly desirable but is rare in these low-to-middle income countries.

Progress in smartphone technology offers a way to build an application for air-conduction-based hearing tests. The advantages of the smartphone-based solution are portability, self-assistance, low cost, and it being less time consuming compared to the formal clinical facilities. Smartphone-based applications can have high reliability, sensitivity, and specificity towards identifying potential hearing loss [14,15,16]. Self-assisted mobile audiometry can provide access and benefits of regular hearing health assessments to a large part of the community, provided accuracy of the hearing thresholds [17]. However, certain challenges such as a wide variation among smartphones, earphone devices, and differences in their specification put a restriction on valid use cases. Only certain combinations of smartphones and earphones can be used to achieve the stated accuracy due to differences in their specification [18]. In addition, smartphone-based audiometry may fail in the instances of severe or asymmetric hearing loss [13].

The study of [19] revealed 30 applications available across iOS and Android platforms that could be used for pure-tone audiometry assessment. Out of these, only five had been through validity studies which include uHear (iOS), shoeBOX audiometry (iOS and tablet), hearScreen (Android), EarTrumpet (iOS), and AudCal (iOS) [19,20]. Except for uHear, these are available on a paid subscription. To the best of the authors’ knowledge, the development and calibration of these applications are not available in the public literature; hence, they cannot be replicated. To have a reliable hearing threshold assessment, it is important to calibrate the smartphone device and the earphones used for smartphone audiometry [14]. Validation results of uHear, shoeBox and EarTrumpet applications are available in [21,22]. The uHear application is well designed for achieving high accuracy for people with hearing loss but it is less accurate for quantifying hearing thresholds for normal-hearing (NH) people [23]. ShoeBox audiometry requires help from a professional assistant to perform the hearing test [24]. HearScreen, EarTrumpet, and AudCal perform best when used for a hearing screening, which does not require providing an accurate assessment of hearing thresholds or an audiogram [19]. Another study [25] proposes a smartphone-based application that requires custom-made hearing aids for testing, which might not be commercially accessible to the major population. The authors of [26] give a web-based solution, which requires third-party assistance from a formal audiometric service professional.

The purpose of this study is to provide an accurate, low-cost, and widely accessible smartphone-based application (HearTest) for quantifying an individual’s hearing ability. HearTest performs pure-tone audiometry (PTA) and plots the test results on an audiogram. In this study, we have used the commercially available iPhone XR smartphone and Sennheiser CX3.00 earphones for calibration and testing purposes. The selected smartphone-earphone combination was calibrated with respect to GSI-61 audiometer and insert earphones (ER-3A), maintaining clinical standards. As part of the calibration, the initial assumption about reference equivalent threshold sound pressure level (RETSPL) values was based on the findings of Ho et al. in [27]. Subjective tests on 20 normal-hearing (NH) people (N=40 ears) were used to correct these RETSPL values. Further subjective testing and retesting on NH participants demonstrated the agreement between the HearTest and the reference audiometer (≤10 dB HL difference). At last, we discuss the limitations and challenges of smartphone-based hearing test application and ways to improve its reliability.

Compared with previous works related to smartphone-based hearing test applications, the major contributions of this paper are as follows:Unlike other smartphone-based screening applications, the proposed smartphone application can quantify the hearing threshold levels within clinically accepted standards for NH people.We propose a novel subjective test-based approach to calibrate a smartphone-earphone combination with respect to the reference audiometer.In PTA, we use continuous tones that are different from regular short interval tones. The advantages of using continuous tones are that they minimize harmonic distortions and do not require the attention of the user.

In the remainder of this text, to minimize the redundancy we define the following terms. By the HearTest application, it is implied that it is used with the combination of the iPhone (8/10/XR) and the Sennheiser CX300 earphone device. A reference audiometer refers to the standard clinical GSI-61 audiometer used with insert earphones (ER-3A). The rest of the paper is organized as follows: Section 2 describes the subjective testing, equipment, and the testing procedure. The methodology of the masking and decision algorithm employed is presented in Section 3. Section 4 gives an idea about the layout of the graphical user interface of the application. The calibration of the HearTest is mentioned in Section 5. Section 6 presents the performance comparison of HearTest to the reference audiometer. Challenges regarding variations in smartphone-earphone combination and further discussions are presented in Section 7. Finally, we conclude the study in Section 8.

## 2. Participants and Equipment

### 2.1. Participants

Subjective tests conducted during this study can be divided into the following four batches based on the HearTest build version (pre-calibration/post-calibration). Application in pre-calibration stage is named “HearTest v1” and the post-calibration stage is “HearTest v2”.

Subjective test batch 1 (B1): For the first version of the application, a total of 20 participants self-identified as having NH ranging from ages 19 to 28 years with mean and standard deviation as 23.88±3.04, except for 2 participants, aged 52 and 56, were recruited. Out of 20, the male and female count was 7 and 13, respectively.Subjective test batch 2 (B2): A total of 14 participants also self-identified as having NH ranging from ages 20 to 30 years with mean and standard deviation as 25.42±3.27 were recruited for a further hearing test with application version 2 (HearTest v2). Out of 14, the male and female count was 4 and 10, respectively.Subjective test batch 3 (B3): five participants from B1 were again called for retesting with the HearTest v2.

Subjects were recruited through a university volunteer research program. All participants were fluent English speakers and had no other reported handicapping conditions. Human subjects’ approval was received from Institutional Review Board of the University of Texas at Dallas, protocol number: 14-72.

### 2.2. Test Procedure and Equipment

During each subjective test, the participant first went through the standard clinical pure-tone audiometry followed by the HearTest application. Both the tests were carried out via air conduction and were performed in an isolated sound booth. For the clinical audiometry, the GSI-61 audiometer and ER-3A insert earphone with specified 50Ω impedance were used. For the HearTest application, iPhone XR was used with Sennheiser CX3.00 earphones, connected through an adaptor. The earphone specified has an impedance of 18Ω and sensitivity of 118 dB SPL/mW.

## 3. Methodology

The standard modified Hughson Westlake approach of down 10 dB, up 5 dB commonly used in audiometric testing was used for quantifying the hearing loss thresholds. Most audiometers have attenuators that are calibrated in 5 dB steps, and more sophisticated models also provide testing in 1 dB, 2 dB, or some other step size that can be manually adjusted. The range of intensities that can be tested with audiometers usually goes from −10 to 120 dB HL. The proposed HearTest application has an upper limit on maximum dB HL that can be tested. It is limited by the maximum output level of the smartphone. HearTest can analyze the hearing loss ranging from −10 to 80 dB HL (except the maximum limit for 0.25 kHz is 75 dB HL).

The application tested six frequencies, 0.25, 0.5, 1, 2, 4, and 8 kHz, for both ears. The test began by presenting the test tone at 30 dB HL. If there was no response from the subject, then the intensity was increased in 10 dB steps until the subject confirmed the test tone to be of sufficient intensity. Once the response was obtained, the test tone was lowered by 10 dB until no response and then was increased in steps of 5 dB. The level that could be heard first after three reversals was assumed to be the hearing threshold level. The responses were recorded on the local memory of the device. Test tones were played continuously so that harmonic distortions were minimum and the subject’s attention did not become an issue [28]. When the threshold for a given frequency was determined, the frequency was changed, and the process was repeated. Testing was performed sequentially for both ears and each frequency mentioned above.

## 4. Smartphone Application

The HearTest application started by asking the user for identification and showed instructions about the test procedure. It asked the user to turn the phone volume to maximum and sit in a quiet location before proceeding towards the test. The graphical user interface of the application is presented in Figure 1. The user was instructed to tap the button corresponding to their response, either “I can hear it” or “I can’t hear it” during the test. The application also monitors the surrounding noise level in the background. If, during the test, background noise level passed above a certain threshold, such as greater than 40 dB SPL, then the test paused in between and notified the user about high surrounding noise and the possibility of inaccurate results. Users could follow the test progress on the progress bar as shown in Figure 1a. Alerts about the changes in the test ear were provided to the user to assist them in understanding the process. It instructs the user to tap the “I can’t hear it” button if they are unsure of their response. The application also allows users to pause/resume the test for sudden interruptions or convenience.

After the successful completion of the hearing test for both ears, the application moved to the pure-tone audiogram page, as shown in Figure 1b. The generated audiogram shows both the right and left ear thresholds on top of each other with similar notations as audiology standards. To assist the user in reading the audiogram, we gave standard stratification of hearing loss as per the American Speech-Language-Hearing Association (ASHA) categorization [29], as shown in Figure 1b. These categories were color-coded and divided as follows. (1) Normal = 0–25 dB HL, (2) mild = 26–40 dB HL, (3) moderate = 41–55 dB HL, (4) moderately severe = 56–70 dB HL, (5) severe = 71–90 dB HL and (6) profound ≥ 90 dB HL. Finally, the application allowed the user to save the test data on the phone or retake the test. The application code is available on GitHub as an open source project [30].

## 5. Calibration

The results of hearing tests are quantified as the hearing level (HL) in decibels for each test frequency. HL is for a given ear at a specified frequency using a transducer, measured with an audiometer calibrated to reference equivalent threshold level (RETSPL) for air conduction [31,32]. For the calibration of audiometric equipment, its RETSPLs should be known to translate output sound level measured in units of dB SPL into dB HL. The relation between them can be expressed as follows,
(1)dBSPL=dBHL+RETSPL.

For example, standard RETSPLs for pure tone when used with calibrated audiometer and insert earphones (ER-3A) are available in ANSI S3.6-2010 and are shown in Table 1. In the following, we talk about how the RETSPL values were estimated for the selected smartphone-earphone combination.

Before building the application, RETSPLs were unknown for selected device combinations, we started with using the best available estimate for RETSPLs. Ho et al. presented RETSPLs for EarPods (MB770G) when used with Apple iPad mini 1 [27]. We used this as our initial best-known RETSPLs for HearTest v1 and are as shown in Table 1. Then, to test the accuracy of the application with assumed RETSPLs, we performed a subjective test on 20 participants from B1 (N = 40 ears) with NH. Subjects underwent air-conduction-based audiometry through a clinical audiometer followed by a hearing test through HearTest v1. Figure 2 shows the mean and standard deviation values of the pure-tone hearing thresholds measured through a clinical audiometer and HearTest v1.

Comparison of the HearTest v1 with clinical audiometry was performed across 0.25, 0.5, 1, 2, 4, and 8 kHz octave frequencies. We can see that measured hearing thresholds with HearTest v1 are not in agreement with clinical audiometry. Spearman’s rank correlation coefficient was found to be 0.14, with a *p*-value greater than 10%. Spearman’s correlation was calculated as data are not normally distributed and might contain outliers. We used this disagreement between the reference audiometer and HearTest v1 to correct the RETSPL values. Due to the linear relation between hearing thresholds and RETSPLs as per (Equation 1), we corrected the initial RETSPLs based on the median of difference in estimated hearing thresholds. The median difference was computed by subtracting hearing thresholds of HearTest from the clinical audiometer, as shown in Figure 3. We rely on median hearing threshold difference instead of mean difference as it was noticed that estimated hearing thresholds did not follow a normal distribution and it has a minimal effect due to outliers. New RETSPL values were incorporated in HearTest v2 and are given in Table 1. Knowing the RETSPLs and test-tone intensity needed to play in dB HL, the intensity of the test tone in dB SPL coming through the transducer connected to the smartphone can be computed using (Equation 1).

## 6. Results

This section compares pure-tone audiometric thresholds for human subjects measured through HearTest v1, HearTest v2, and clinical audiometry. Comparison is among six octave frequencies—0.25, 0.5, 1, 2, 4, and 8 kHz. HearTest v2 has corrected RETSPL values and since we used the audiometer as our reference in calibration, we expect it to perform similarly to the audiometer. We carried out subjective tests on 14 NH participants (N=28 ears) from B2 and plotted the mean and standard deviations of the measured hearing threshold in Figure 4. Here, we could observe the difference in mean values between HearTest v2 and the audiometer has reduced significantly compared to HearTest v1. The maximum difference of mean values among all the six frequencies was 4.64 dB HL, which is lower than the clinical test–retest limit of 10 dB HL. Spearman’s rank correlation coefficient was found as 0.92 with a *p*-value of less than 1%. This validates the efficiency of the proposed calibration method.

To ensure the reliability of the calibrated application, we retested the five participants (N=10 ears) from batch B1 with HearTest v2. Figure 5 shows the mean absolute error (MAE) in hearing thresholds when measured with HearTest v1, HearTest v2, and compared with a clinical audiometer. Mean error in hearing thresholds for the same participants was higher in HearTest v1, which improved significantly later in HearTest v2. Maximum MAE of 14- and 5.5-dB HL was observed for HearTest v1 and v2, respectively, when compared with clinical audiometry.

## 7. Discussion

There are two critical concerns of variability in choosing devices; a smartphone and an earphone. In 2017, a study [33] the maximum difference of 8 dB in the sound output level among six smartphone devices distributed by four major smartphone manufacturers (Apple, Samsung, LG and Nokia). For that reason, a valid hearing test can be conducted only on the devices that are well-calibrated with the clinical audiometer, which limits the accessibility of the smartphone-based audiometry. RETSPLs for ten Android-based smartphone devices with bundled earphones were investigated in [34], which found no significant difference when devices shared a similar brand. To test this with Apple devices, we measure the maximum output level of different Apple devices using a 2-cc coupler and are as shown in Table 2. The maximum difference of 2 dB SPL was noticed, which is relatively low. This suggests that hearing tests performed through Apple smartphone devices reduce variability and can be reliable. The proposed calibration method can be applied to other device platforms too such as Android. The variations in the earphone impedance can affect the hearing test accuracy. To minimize this effect, we recommend using earphones with similar specifications as this study. In future work, we may focus on deriving the calibration values of one smartphone-earphone assembly with respect to another by extracting the device specifications such as smartphone impedance, maximum output level, earphone impedance, sensitivity, etc. This can substantially reduce the subjective testing needed for calibration.

Another limitation posed in smartphone-based audiometry is “crossover”, a phenomenon where sound presented to one ear can be heard by the other ear. The interested reader can find deeper insights at [35,36]. Crossover can misconstrue the user’s hearing thresholds in testing, particularly in patients with an asymmetric hearing loss. In brief, to mask the air-conduction of sound from the test ear, narrow-band noise centered around the test frequency is played to the non-test ear (NTE). This is called masking, which is carried out so the tone cannot be heard in the NTE in order to obtain a true response from the test ear. It might seem logical that sound escaping through the foam of earphones or traversing through the external path has a greater impact on the crossover. However, it has been shown in research that the actual crossover route for air conduction signals occurs principally by bone-conduction to the cochlea of the NTE [37,38]. When a signal “crossover” happens, the amount of dB “lost” in air-conduction is called “interaural attenuation” (IA). Average IA values are approximately 50–65 dB for standard supra-aural headphones (TDH 39). However, we cannot rely on the average values. The minimum suggestion for IA for crossover in clinical purposes is 40 dB [35]. This may limit the maximum dB HL range for a smartphone-based hearing test. For example, if a tone of 60 dB SPL is played to the TE, it is possible that 20 dB may arrive at the NTE. Therefore, a masking tone of at least 20 dB SPL would be needed in the NTE. When enabling a hearing test for people with hearing impairedness (HI), it is necessary to know the IA of the earphone used.

In future work, we plan to perform subjective testing on people with HI to validate the HearTest application for people with mild to severe hearing loss. We have incorporated the masking feature in the new version of the application, which will be necessary when testing people with HI, who typically have high hearing thresholds. To study the importance and relevance of masking, the application might be tested on participants with asymmetric hearing loss. To improve the accuracy of the proposed application, the threshold detection algorithm can be fine-tuned further with 2 dB HL down and 1 dB HL steps. The HearTest application is currently designed for adults and can be improved to be accessible to children. Assessment of the application needs to be quantified when used in quiet room that replicates the practical scenario. Furthermore, it is important to note that, although the proposed smartphone-based application has its advantages and limitations, it can not substitute for a professional examination by an Audiologist.

## 8. Conclusions

This study presents an approach to develop a smartphone-based hearing test application, which can accurately and reliably quantify the hearing threshold levels for NH people when compared with clinical audiometry. The proposed application followed the standard audiometric approach of 10 dB down and 5 dB up for identifying the hearing thresholds. Calibration of the smartphone-earphone pair (iPhone XR with Sennheiser cx3.00 earphones) was carried with respect to the clinical audiometer (GSI-61 with insert earphones ER-3A) through subjective testing on 20 NH participants. Evaluation of the calibrated version of the application with 14 NH participants showed a maximum 4.6 dB HL of mean absolute difference when compared with the reference audiometer, which is significantly lower than the clinically accepted standard (10 dB HL). Retesting of five NH participants before and after calibration supported the validity of the calibration. Relatively low variation of maximum output level among devices of the same model (Apple) suggested that it is feasible to carry out the hearing test using the same calibrated RETSPL values until earphones of the specified impedance and sensitivity are used. Future steps on validation of the application through subjective testing on people with HI and challenges due to “crossover” phenomena were discussed.

## Figures and Tables

**Figure 1 ijerph-18-05529-f001:**
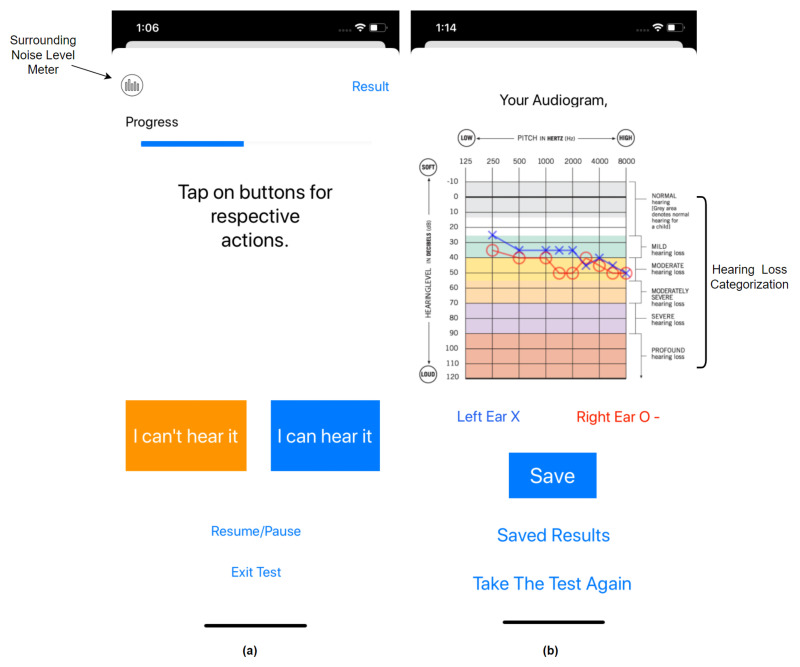
Graphical user interface of HearTest application on iPhone XR: (**a**) test page where the response was recorded (**b**); audiogram plot for the artificial hearing test.

**Figure 2 ijerph-18-05529-f002:**
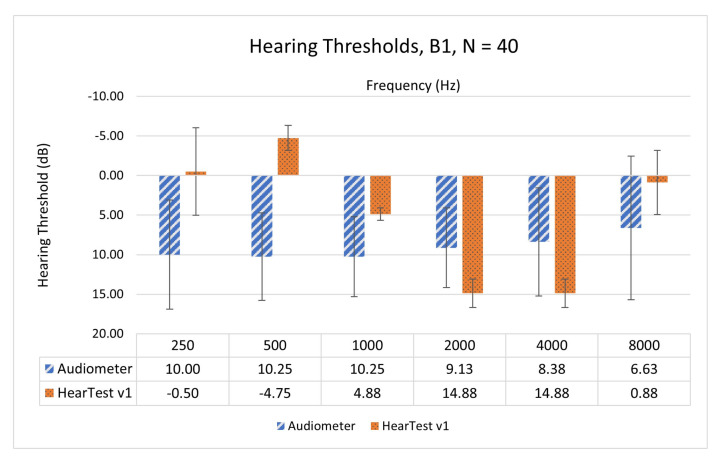
Mean and standard deviations of hearing thresholds using Audiometer and HearTest v1 on NH people from B1 (N=40).

**Figure 3 ijerph-18-05529-f003:**
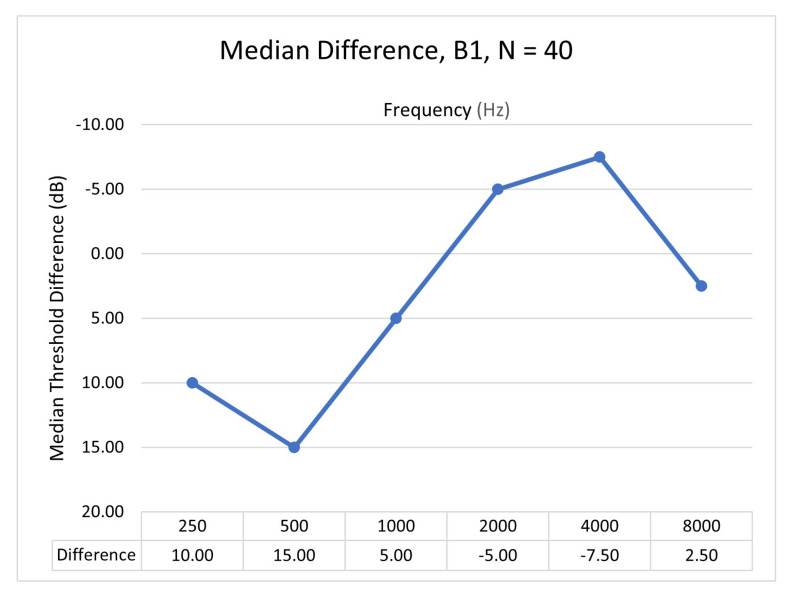
Median difference of hearing thresholds using Audiometer and HearTest v1 on NH people from B1 (N=40).

**Figure 4 ijerph-18-05529-f004:**
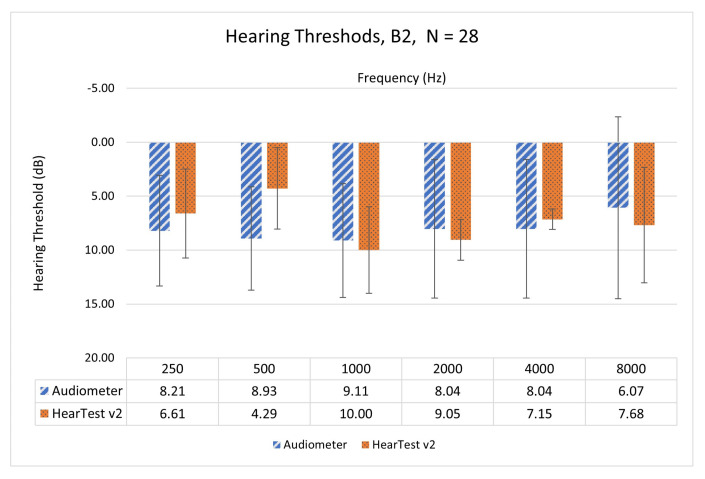
Mean and standard deviation of hearing thresholds using Audiometer and HearTest v2 on NH people from B2 (N=28).

**Figure 5 ijerph-18-05529-f005:**
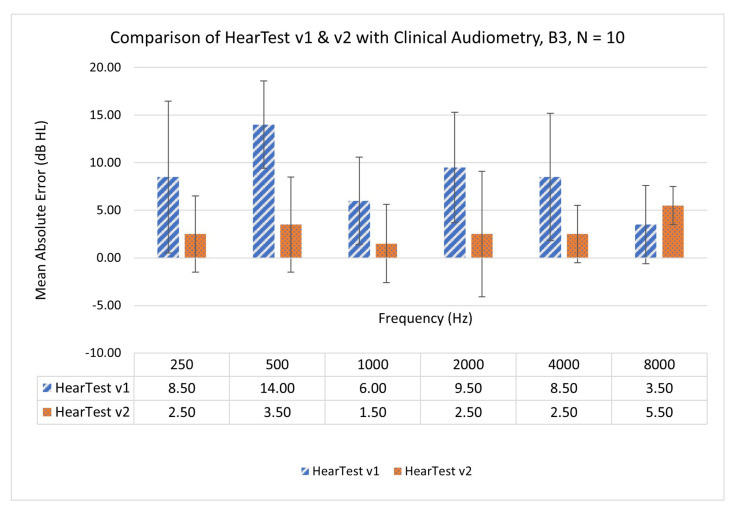
Retest on five people (N=10 ears) from B1 with app v2. Mean and standard deviation of absolute error in hearing threshold measured for HearTest v1 and v2 when compared with clinical audiometry.

**Table 1 ijerph-18-05529-t001:** Reference Equivalent Threshold Sound Pressure Level (RETSPL).

	250	500	1000	2000	4000	8000
ER 3A	14.5	6.0	0	2.5	0	−3.5
HearTest v1	11.8	9.5	6	2.3	6.7	16.2
HearTest v2	1.8	−5.5	1	7.3	13.7	13.7

ER-3A: ANSI S3.6-2010, American National Standard Specification for Audiometers, Acoustical Society of America, HearTest v1: RETSPLs for Apple Earpods, Ho et al. [27], HearTest v2: calibrated using subjective test model based on ER-3A and HearTest v1.

**Table 2 ijerph-18-05529-t002:** Maximum output in dB SPL measured with 2-cc coupler for mentioned frequencies (in Hz) played through Sennheiser CX300 earphone connected across different iPhone devices.

Smartphone	Channel	250	500	1000	2000	4000	8000
iPhone XR	L	85.1	93.5	107.2	112.8	106	108.8
	R	84.9	93.4	107.3	113	106.1	108.7
iPhone 10	L	83.4	94.6	107.8	113.5	105.7	108.6
	R	83.5	94.1	108.1	113.2	105.8	108.2
iPhone 8	L	83.1	93.3	107.3	112.8	106.4	108.5
	R	83.3	93.3	107.2	112.7	106.1	108.9

## Data Availability

The data presented in this study are available upon request from the corresponding author. The data are not publicly available in order to protect personal health information regarding hearing abilities. Participants gave consent to the publication of the results provided that coding system would be used to protect identity.

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
