# Peer review of "Development and Pilot Testing of Smartphone-Based Hearing Test Application"

_ijerph, 2021, doi:10.3390/ijerph18115529_

Round 1
Reviewer 1 Report
This article deals with the validity and reliability of smartphone -based application for assessment of our hearing ability. The application will provide us the accurate, easy assessment method of hearing ability. Furthermore the development will contribute to the improvement of hearing screening rate. The study design, procedure, data analysis, and conclusions appropriate. The following minor revisions may be needed to improve the quality of the article.
Minor revision suggestions:
- Could you show the mean and standard deviation of the study participant’s age? (Line125-, 129-) The authors only showed age range of them. Additionally, the details of subjects participating in batch 3 is not shown.
- Authors wrote “Subjects with hearing loss were recruited from previous experiment lists.” (Line 135) What test is these subjects with hearing loss targeted for?
- The statistical analysis is needed to compare the hearing thresholds using Audiometer and HearTest. (Figure 2, 4, 5) If authors can show the results using statistical analysis (ex. correlation coefficient), the study findings will be more objective and accurate ones.
Reviewer 2 Report
Overall
The organization of this manuscript is concise and logical. The topic is interesting for the development of teleaudiology, especially during a pandemic. However, the sample size is not age-diverse, the number of participants is small, and the study was conducted on only one cell phone model. We suggest renaming the title to pilot study.
Section 1. Introduction
The literature review could be enriched with more detailed results of hearing tests performed using smartphones, including a discussion of several key studies and mention of comprehensive reviews of research in this area. In addition, the authors could consider expanding the literature review to include information from previous work describing the benefits of early identification of hearing loss using smartphone apps and/or teleaudiology solutions.
The importance of continued monitoring of hearing sensitivity is an important topic. Prior to reading this manuscript, it may not be evident to all readers that hearing examination required the phone model and headphones specified by the author.
In line 48-49 authors indicated, that most people with hearing loss live in low-income countries. To my knowledge, an Apple-grade phone is not readily available to this audience. Has testing been attempted on other phone models ? Please clarify your choice of the iPhone.
- 67-68; ‘Only certain combinations of smartphones and earphones can be used to achieve the stated accuracy’ Please clarify and add more citation.
Section 2.1. Participant
Did all subjects have a confirmed hearing normal when tested on a clinical audiometer? In case of hearing impairment was the volunteer excluded from study?
Why aren't the B1 and B2 groups composed of the same individuals who had the test done pre- and postcalibration? Please add more details.
Section 2.2. Test Procedure and Equipment
Have you considered performing pure tone audiometry using the HearTest application in a quiet room environment rather than in an isolated sound booth? Or making a third measurement that simulated the acoustic conditions of random users?
Section 3. Methodology
l.163-165; ‘Test tones are played continuously so that harmonic distortions are minimum, and the subject’s attention does not become an issue’ - Consider add citation
Section 7. Discussion
How does the authors' proposed calibration relate to the biological calibration in HearTest that is performed each time the application is launched?
However, it is important to be aware of possible risks (e.g. safety, quality, efficacy, privacy), and for patients and professionals to be aware of the safe use of apps. It is also important to remember that apps are only for a visual assessment of hearing and are not a substitute for a professional examination by an Audiologist.
